# A Hitchhiker's Guide to Scaling Law Estimation

## Abstract

Scaling laws predict the loss of a target machine learning model by extrapolating from easier-to-train models with fewer parameters or smaller training sets. This provides an efficient way for practitioners and researchers alike to compare pre-training decisions involving optimizers, datasets, and model architectures. Despite the widespread use of scaling laws to model the dynamics of language model training, there has been little work on understanding how to best estimate and interpret them. We collect (and release) a large-scale dataset containing losses and downstream evaluations for 485 previously published pretrained models. We use these to estimate more than 1000 scaling laws, then derive a set of best practices for estimating scaling laws in new model families. We find that fitting scaling laws to intermediate checkpoints of training runs (and not just their final losses) substantially improves accuracy, and that—all else equal—estimates of performance are generally most accurate when derived from other models of similar sizes. However, because there is a significant degree of variability across model seeds, training multiple small models is sometimes more useful than training a single large one. Moreover, while different model families differ in scaling behavior, they are often similar enough that a target model's behavior can be predicted from a single model with the same architecture, along with scaling parameter estimates derived from other model families.

## 1 Introduction

Substantial effort and cost are required to train even a single large language model (LLM).[1] There is thus an acute need for efficient decision-making aids that can evaluate the effectiveness of proposed changes to language models' architecture or training data without full-scale training runs. While there is a large body of work that motivates or evaluates these changes using small models (Warstadt et al., 2023; Hillier et al., 2024), synthetic tasks (Akyürek et al., 2024; Wortsman et al., 2023) or theory (Jelassi et al., 2024), one of the most important tools for current practitioners is the estimation of **scaling laws** for LLMs (Ivgi et al., 2022; Dubey et al., 2024).

A scaling law extrapolates the performance of a target model from the performance of a set of models with fewer parameters or smaller training sets. Typically, this extrapolation requires models to belong to the same **model family**, differing only in parameter count and training set size, but using the same architecture and training distribution. A high-quality scaling law accurately predicts the target model's test performance (Rosenfeld et al.; Kaplan et al., 2020; Hoffmann et al., 2022).

Most past work describing and characterizing scaling laws has begun by exhaustively training models in a family across a full range of dataset sizes and parameter counts. One question that has received comparatively little attention is how, when training a new LLM, a practitioner with limited computational resources should choose *which* small-scale models to train in order to best estimate a target model's final performance. This paper offers a practical guide to when, and how, to use small models to efficiently obtain meaningful predictions about large models' behavior—maximizing prediction reliability while minimizing the budget for preliminary experimentation, which necessarily involves tradeoffs between the number of preliminary models trained, the size of the largest preliminary model, and size of the dataset used to train it.

---

[1]Code, data and full numbers are found in our repository

We begin by collecting diverse model data to perform a large-scale meta-analysis of scaling laws (§3). Usually, scaling law research relies on a single collection of closely related models, or alters only a minimal aspect of pretraining (e.g. data size; Muennighoff et al., 2024). Instead, we gather data from as diverse a set of scaled families as possible, to allow this and future meta-analysis of scaling laws that generalize across architectures, datasets and settings.

The rest of the paper uses this data to analyze a number of key questions around scaling law estimation:

1. **How reliably may we expect scaling laws to extrapolate?** Variation between random parameter initializations can produce changes of up to 4% in loss. Most published improvements in pretraining procedures, when performing minimal controlled experiments, report loss changes between 4% and 50% (§4).

2. **How much does the shape of scaling laws vary across model families?** Different model families have scaling laws with a different functional dependence on model size (§5). However, transformer LMs are similar enough that, with a single model from a target family and a scaling law from a different model family, it is sometimes possible to accurately estimate target model performance (§5.1).

3. **Must scaling laws be estimated only from fully trained models?** Even though optimization procedures are typically sensitive to the full size of a training run, estimating scaling laws from intermediate training checkpoints greatly improves scaling law fit (§6). It is generally possible to estimate a model's final loss beginning roughly $1/3$ of the way through training.

4. **How large must models be to produce reliable scaling estimates?** All else equal, experimenting with large models is typically more useful than with small models (§7), but may be outweighed by the benefits of reduced variance from training more, smaller models (§8).

5. Taken together, **cost-effective estimation** of a scaling law should consider the **number** of models, the **size** of the models, and the number of **training tokens** for each model. We highlight those size, tokens and number of models effects in Fig. 2.

Our experiments also provide insight into the functional form of scaling laws themselves, suggesting that they may have fewer degrees of freedom (§9) than typically assumed. We conclude with discussion of other work on scaling law estimation that may be of interest to practitioners §10.

## 2 DEFINING A SCALING LAW

A scaling law estimates the loss of a costly model by training cheaper ones (see Fig. 1) which share a pretraining procedure and differ by some hyperparameters, typically model size (#params) and number of tokens seen during training (#toks). A scaling law is a function that predicts a target model's loss on held-out data when setting the value of one hyperparameter (Kaplan et al., 2020) or both (Rosenfeld et al.; Hoffmann et al., 2022). Comparing laws' predictions about different pretraining choices (e.g. data Ge et al., 2024) allows informed decisions about which large-scale model to train.

A scaling law also enables finding the optimal choice of hyperparameters under computational constraints on pretraining (Hoffmann et al., 2022) or inference (Touvron et al., 2023; Sardana et al.).

Formally, we will call a **model** $f$ any single concrete neural language model with a specific set of parameters. Different seeds, or even different checkpoints from the same training run, correspond to different models. We define a **scaled model family** $f$ as a set of models, with each $f \in F$ differing only in size #params($f$) and number of tokens #toks($f$).

There are two specific subsets of scaled model families that will be useful in our experiments. First, the **maximal parameter family** $\max_{\#\text{params}}(F)$ contains only models in $F$ with the largest number of parameters. Formally, define $m = \max_{f \in F}$ #params($f$); then $\max_{\#}$params($F$) =

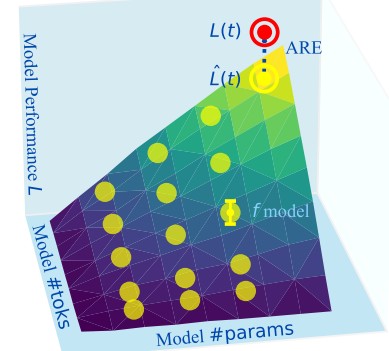

Figure 1: Illustration of a scaled family, an estimated scaling law, and its prediction error for a target model.

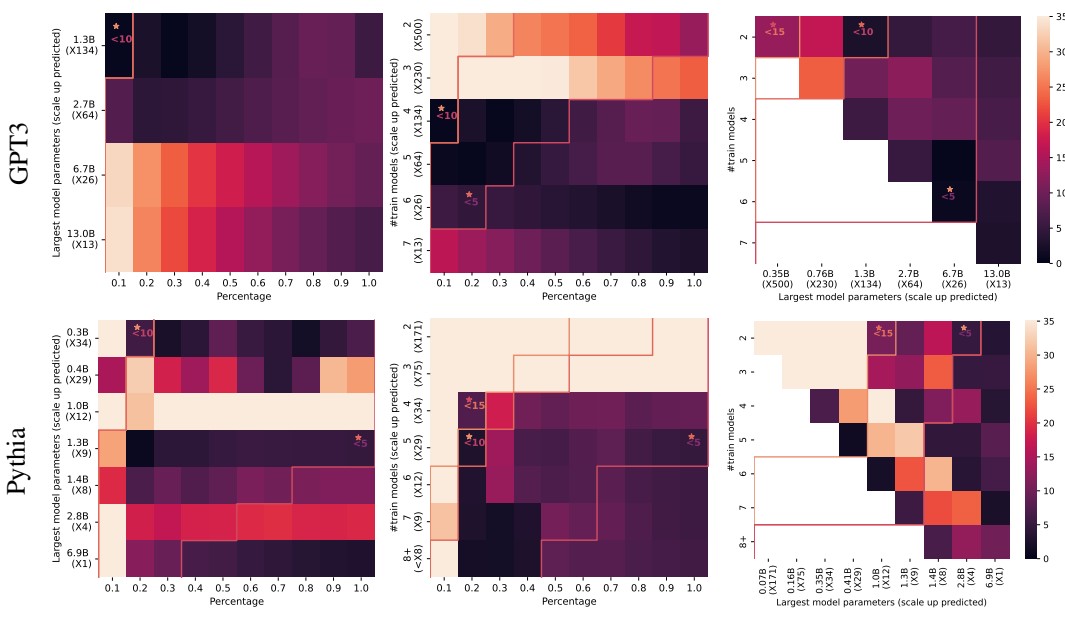

(a) Scale up vs. Train Percentage    (b) #Models vs. Train Percentage    (c) #Models vs. Scale up predicted

Figure 2: The effects of three variables on scaling law accuracy. Each cell corresponds to a single scaling law estimated from a set $F_{\text{train}}$ of model checkpoints, with the color denoting that scaling law's error when predicting the largest model in a family. Each column shows a subset of the three axes along which these training sets differ: (1) the number of tokens used to train each LM in $F_{\text{train}}$ (expressed as a fraction of the full training corpus), (2) the number of distinct models trained; and (3) the size of the largest model trained (expressed as a scale-up factor—the ratio between the target model and the largest model in $F_{\text{train}}$). In (a), all laws are estimated from four models. In (c) all laws use the full corpus. Orange lines show iso-FLOP contours (sets of scaling laws whose training sets require the same computational cost to produce). ★ represent the most efficient ways to obtain 15%, 10% and 5% ARE. One of the most immediate conclusions from these plots is that scaling law estimation is quite noisy—the inclusion of a single badly-behaved model in the estimation procedure can produce large errors, and in small model families error does not reliably decrease with additional computation. However—because of noise—it is often preferable to extrapolate from a large number of small, partially trained models rather than a small number of large models.

$\{f \in F \; : \; \#\mathsf{params}(f) = m\}$. This family will generally contain the **target** model(s) whose behavior we wish to predict $t \in F_{\text{target}}$. Second, the **q-maximal token family** $\max_{\#\mathsf{toks}}(F, q)$ contains all models in $f$ trained on at least a $q$-sized fraction of the training set. Formally, define $t = q \cdot (\max_{f \in F} \#\mathsf{toks}(f))$; then $\max_{\#\mathsf{toks}}(F, q) = \{f \in F \; : \; \#\mathsf{toks}(f) \geq t\}$. Note that this definition does not distinguish between partially trained models on one hand, and models trained to convergence on a subset of the largest training set used in a family on the other. Throughout this paper, we will not in general distinguish between these two types of models, a decision evaluated in Section 6. Indeed, except where noted, $\max_{\#\mathsf{toks}}(F, q)$ should be thought of as containing the checkpoints from the last $q\%$ of a training run.

A **scaling law** $\hat{L}(f \mid F)$ estimates the performance of a new model $f$ given a model family $F$. (We will simply write $\hat{L}(f)$ when the family is clear from the context.) All experiments in this paper use the common functional form from the literature (Hoffmann et al., 2022):

$$\hat{L}(f) := E + \frac{A}{\#\mathsf{params}(f)^\alpha} + \frac{B}{\#\mathsf{toks}(f)^\beta}. \tag{1}$$

Here $E$ captures the scaled family's general performance; $A, \alpha$ and $B, \beta$ describe the scaling effect of $\#\mathsf{params}$ and $\#\mathsf{toks}$ respectively.[2] These parameters are estimated by first collecting a set of

---

[2]We believe many of the findings in this paper apply to other functional forms that have been proposed for scaling laws (Caballero et al.), and even suggest new parameterizations as described in §9.

**training models** $F_{\text{train}}$, then minimizing the reconstruction error

$$\underset{E,A,\alpha,B,\beta}{\arg\min} \sum_{f \in F_{\text{train}}} (\hat{L}(f) - L(f))^2$$

where $L(f)$ denotes the empirical negative log-likelihood of some held-out data under the model $f$.

In this sense, a scaling law is an ordinary parametric machine learning model, and we may ask many of the same questions about $\hat{L}$ that we ordinarily ask about LLMs —what training data ($F_{\text{train}}$) should we collect? How do we estimate accuracy? We seek to provide empirical answers to these questions, for which we first require data.

## 3 DATA FOR 1000+ SCALING LAWS AND MORE

As part of this work, we have collected and released the largest-scale public dataset describing scaling behavior across model families. This dataset aggregates information from a large number of LLM training efforts that have released information about the behavior of multiple models of different sizes or scales. While experiments in this paper focus on scaling laws that measure loss, the dataset also includes information about model performance on downstream evaluation benchmarks where available. We have focused on language models where the largest one is more than 3B parameters and where data was shared publicly or in private correspondence. Our repository accepts further contributions and requests for additions. In addition to those, we have manually extracted some data from papers that did not release models but reported losses in figures.

### 3.1 DATA SOURCES

For each model in this dataset, we report any downstream evaluation and loss that was measured during training, as well as calculated #toks for each, links to matching checkpoints when available, links to data sources, and information about computational cost (in FLOPs) and number of training epochs (i.e. passes over the training set). Each model is identified by a unique name, a type (e.g. llama), #toks, #params, architecture type (e.g. encoder-decoder), and seed.

Models in this dataset include Pythia (Biderman et al., 2023, which provides the largest set of models and variations in a family), OPT (Zhang et al., 2022, collected thanks to Xia et al., 2023; Biderman et al., 2023), OLMO (Groeneveld et al., 2024), Amber (Liu et al., 2023), K2 (Team, 2024), Mamba (Liu et al., 2023) RedPajamas [3] ModuleFormer mixture of experts (Shen et al., 2023), overtrained models (Gadre et al., 2024), Mamba, Llama and hybrid architecture variations from Poli et al. (2024), transformer architectures (Alabdulmohsin et al., 2022), Bloom (Le Scao et al., 2023), T5-Pile (Sutawika et al., 2024), Pandey (2024) models, GPT-family models with different data regimes (Muennighoff et al., 2024), Gopher (Hoffmann et al., 2022) and GPT3 (Brown et al., 2020).

The data consists of 1.9M steps of training evaluated on loss or perplexity, usually on multiple data sources belonging to 485 unique pretrained models, and more than 40 scaled families. To combine the data together, each loss or score was kept in its original form and an aggregating function to normalize the scores to similar scales (perplexity and loss have a log relation) and average was supplied. The data, saved as a compressed table with all the metadata and scores together is hence large but comprehensive.

We hope this will provide a useful resource for the community and plan to extend it further as models get released and their training dynamics are shared. We see such a resource as a facilitator to more research on model development (e.g. A/B testing), scaling laws, downstream scaling laws (Gadre et al., 2024; Ruan et al., 2024; Owen, 2024; Isik et al., 2024), training dynamics (Choshen et al., 2022) and more.

### 3.2 SCALING LAW ESTIMATION

In the rest of the paper, we present findings from estimating hundreds of scaling laws as follows:

---

[3]https://www.together.ai/blog/redpajama-models-v1

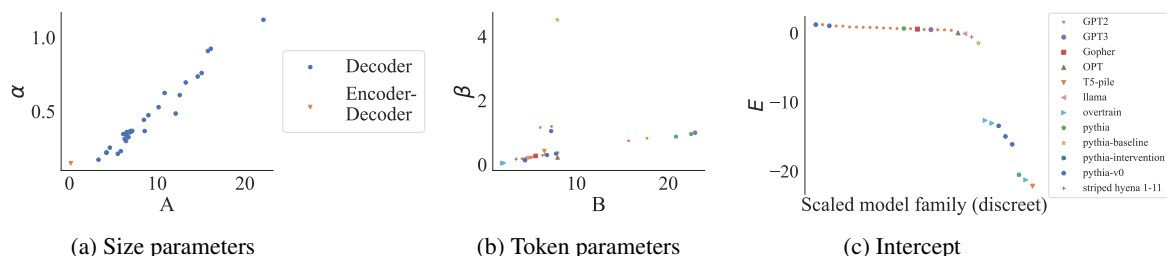

(a) Size parameters  (b) Token parameters  (c) Intercept

Figure 3: Parameters differ between scaled model families. Surprisingly, however, the pairs of parameters controlling the influence of model and training set size have similar ratios. The legend shows model architecture (left), scaling families (center) and per-family intercept (right).

**Fitting**  For each model family $F$, we identify the maximal parameter family $F_{\max} = \max_{\#\text{params}}(F)$, and estimate a scaling law $\hat{L}$ using the remaining models $F_{\text{train}} = F \setminus F_{\max}$. Estimation of scaling law parameters uses the `curve_fit` function in `scikit-learn` (Pedregosa et al., 2011). We additionally experimented with an L-BFGS-based solver but found it to be less stable. We only estimate scaling laws for model families that contain at least three models.

**Evaluation**  To evaluate estimated scaling laws reliably, we need to account for loss fluctuations during large-scale model training. Thus, we test against a few checkpoints near the end of training: we choose as target models $F_{\text{target}}$ the 30%-maximal token family from the set $F_{\max}$ defined in the previous paragraph—that is, we take $F_{\text{target}} = \max_{\#\text{toks}}(F_{\max}, 0.3)$. We then report the mean **absolute relative error (ARE)** $\mathbb{E}_{f \in F_{\text{target}}} |L(f) - \hat{L}(f \mid F_{\text{train}})|/L(f)$ between the empirical loss $L$ and the loss $\hat{L}$ predicted by the scaling law. We reran our experiments with huber loss see §E.

# 4 HOW WELL CAN I EXPECT A SCALING LAW TO PREDICT?

*4% is the best ARE typically obtained; ARE up to 20% can still distinguish between many modeling choices.*

To establish how accurate a scaling law must be to be *useful* to practitioners, we first assess what changes in model accuracy have been considered meaningful in past work. We have surveyed experiments in the literature where an A/B test was performed, i.e., two models were trained similarly, manipulating one attribute to see how it affects scores. Empirically, we found no widely adopted modeling changes that were motivated with less than a 4% relative difference between models. Additionally, reported variance across random restarts of the same model architecture reaches up to 3.5% (c.f.,§8; Sellam et al., 2021). We take this to mean that this is approximately the minimal effect-size experimenters care about and possibly the minimal effect one can reliably measure. Accordingly, this bounds the best goodness of fit we should expect or require of scaling laws.

To offer several concrete points of comparison: Pythia 6.9B models fixed inconsistencies in their code and hence have two versions (c.f. App. B; Biderman et al., 2023) which differ in loss by 40%. They also provide data deduplication A/B test that had a minor effect on the loss of about 5%. Gadre et al. (2024) tested the effect of training 400M parameter models for different #toks. The most similar (double the training tokens) has approximately 4% change and can reach a 50% loss difference with 30 times more training. Training on a constant #toks but repeating the same data resulted in almost no changes for up to 4 repetitions (epochs), and later in about 8%, 50% on 14.44 repetitions of the data (Muennighoff et al., 2024). Instead of varying the amount of data or epochs, Ge et al. (2024) found that training on a different kind of data incurred ARE of approximately 10% and different data mixes led to 6% changes or less.

## 5 WHEN I TRAIN A NEW MODEL, DO I EVEN NEED A NEW SCALING LAW?

*Different model families exhibit different scaling behavior, but performance can sometimes be estimated using a single model in a new family.*

Scaling laws relate performance to scalar training parameters like model or dataset size. For discrete decisions (whether the choice of nonlinearity or data preprocessing scheme), it is not immediately obvious how to pool information across models that differ in these traits (see Ruan et al., 2024; Maia Polo et al., 2024, for concurrent work that performs this pooling based on downstream task behavior). Clearly, different pretrained models with the same #params and #toks still show different loss, so these differences can be consequential. But how do discrete choices of architecture, training procedure, or dataset, affect the form of scaling laws?

One way to answer this question is to look at the parameter estimates for scaling law parameters $E$, $\alpha$, $A$, $\beta$ and $B$ differ across model families. These results are shown in Fig. 3, where it can be seen that there are often dramatic differences in all five parameters across families. In this sense, even the rate at which additional data or parameters improve model performance depend on underlying architectural details, suggesting that understanding the behavior of a new model family may require a new scaling law.

But another way to answer this question is to ask how reliably we can predict final model accuracy when borrowing (or pooling) some parameters of scaling laws between families—even if these result in poor parameter estimates, they may predict large-scale model behavior within the range of meaningful differences identified in Section 4. To do so, we set the #params scaling parameters $(A, \alpha)$ to fixed values reported in past work, and estimate remaining parameters for individual model families. We take the variable values found by Muennighoff et al. (2024) (see Besiroglu et al., 2024; Porian et al., 2024 for a discussion of estimates from earlier work including Hoffmann et al., 2022). We find (see Fig. 6 in App. A) that in some cases only a single training run in a new model family is necessary to obtain accurate scaling law predictions. In the OLMO family, for example, we obtain less than 1% error estimating the accuracy of a 7B model from a collection of 1B model checkpoints. We find that predictions generalize, and a constant #params scaling factor is enough for most models (except the encoder-decoder T5-Pile). However, error rates are larger than in the source family, and predictions for larger models are worse (most conspicuous in OPT's error of 37%, 25% and 15% when extrapolating from 8.7B, 13B and 30B to 175B).

### 5.1 CAN I JUST TRAIN THE TARGET MODEL A BIT INSTEAD OF MANY SMALL MODELS?

*Yes, but obtaining reliable estimates in this way requires up to 30% of the full training run.*

The above results (last row of Fig. 6 in App. A) also suggest the possibility of predicting losses not with just smaller models, but with partially trained versions of the target model itself. When predicting inside the same #params family—that is, estimating $\hat{L}(f \mid F_{\text{target}} \setminus \{f\})$— the #params term in Eq. (1) is constant, and extrapolation is only required for #toks. As seen in the figures, this form of estimation is informative if permitted by computational constraints. Beyond the immediate usefulness of this approach, it is a promising avenue for future research. Better adjusting the scaling laws for predicting through training might improve this efficiency.

### 5.2 ARE EVEN SIMPLER BASELINES ENOUGH?

*Some extrapolation is necessary: scaling laws can produce accurate estimates even when the target model vastly outperforms any training model.*

To provide another form of comparison for the predicted scaling laws, we compute two baselines. Both baselines adopt a pessimistic evaluation assuming that the target model is no better than the best model in the small model family used to estimate a scaling law. Specifically, the baselines are the *best performance* $\hat{L}(\varnothing \mid F_{\text{train}}) = \min_{f \in F_{\text{train}}} L(f)$ and the performance of the *most-trained model*, consuming the most compute for training, i.e. $\hat{L}(\varnothing \mid F_{\text{train}}) = \arg\max_{f \in F_{\text{train}}} \#\text{params}(f) \times \#\text{toks}(f)$. Those baselines might be the best one can expect without fitting a law to scaling.

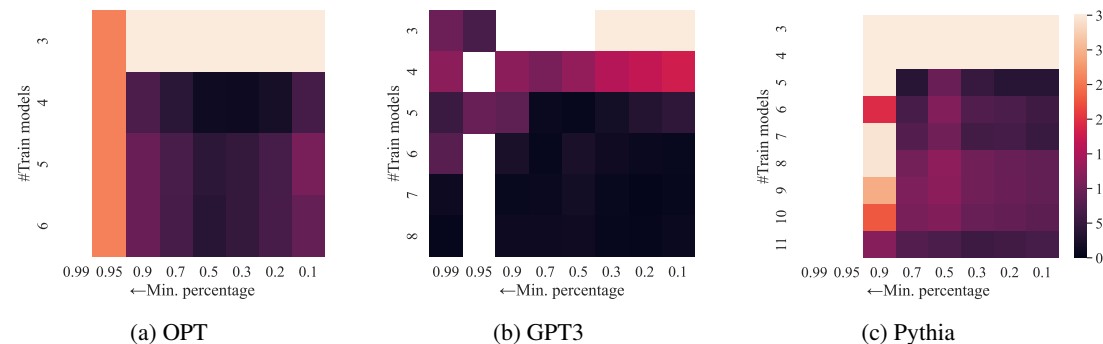

(a) OPT                (b) GPT3                (c) Pythia

Figure 4: The effect of fitting on more of the training trajectory. Each cell represents the absolute relative error estimating scaling laws from a given number of models (vertical axis) trained on a given subset of the *final* checkpoints from a training run (so scaling laws on the left are estimated using all checkpoint, and laws on the right are estimated using only the final 10% of checkpoints). White cells failed to fit. As long as the first ≈10% of checkpoints are discarded, final loss can often be predicted accurately.

We find (See App. 5.2) that out of the two, the *best performance* baseline is closer to $L(F_{\text{target}})$, which is to be expected, as the target model performance is better than any other model in $F$ and this is the better of the two. In both cases, even with the full $F$, the baselines suffer more than 15% error, mostly above 10%, almost never get below 5%, and 18% ARE on average across all scaled families we study.

## 6   I HAVE SOME DATA, WHAT PORTIONS SHOULD I USE?

> *Estimate scaling laws from intermediate checkpoints, not just fully trained models!*

Most past work on scaling behavior of language models (e.g., Gadre et al., 2024; Muennighoff et al., 2024) has trained a *separate* model for each value of #toks studied. This is based on the assumption that changes in the learning rate schedule, which depend on the size of the full dataset that will be used for training, render losses from intermediate checkpoints uninformative.

However, some recent work has demonstrated the effectiveness of learning schedules that do not require prior access to the size of the training set (Hu et al., 2024), and some work has questioned whether careful choice of the learning rate decay is even necessary for reliable scaling laws (Porian et al., 2024). Together, these findings motivate revisiting the assumption that only a single useful datapoint may be obtained from each training run. In the final portion of §5.1, we observed the value of intermediate checkpoints when only a single #params family is used to fit a scaling law. Mathematically, there has to be some difference between models trained on the same number of tokens and sizes depending on the scheduler, it is unknown however, how big is this difference. We now test whether this finding extends to larger families—i.e. whether including intermediate checkpoints from all models in a model family reduces ARE.

Results are shown in Fig. 4, which plots ARE for scaling laws estimated from data subsets of the form $\max_{\#\text{toks}}(F, q)$ for varying $q$. We find that including full training curves in scaling law estimation can predict losses well. In fact, relying merely on the end of training produces significantly worse performance across the board. Our remaining experiments thus fit scaling laws using all these intermediate checkpoints, and not final performance alone.

### 6.1   SHOULD I USE ALL INTERMEDIATE CHECKPOINTS?

> *Almost all, but drop checkpoints from the beginning of training.*

In Fig. 4, we plot the ARE for different **q**-maximal token families serving as $F$, i.e., when fitting only with the end of training runs. There is not a clear trend indicating whether we should use all

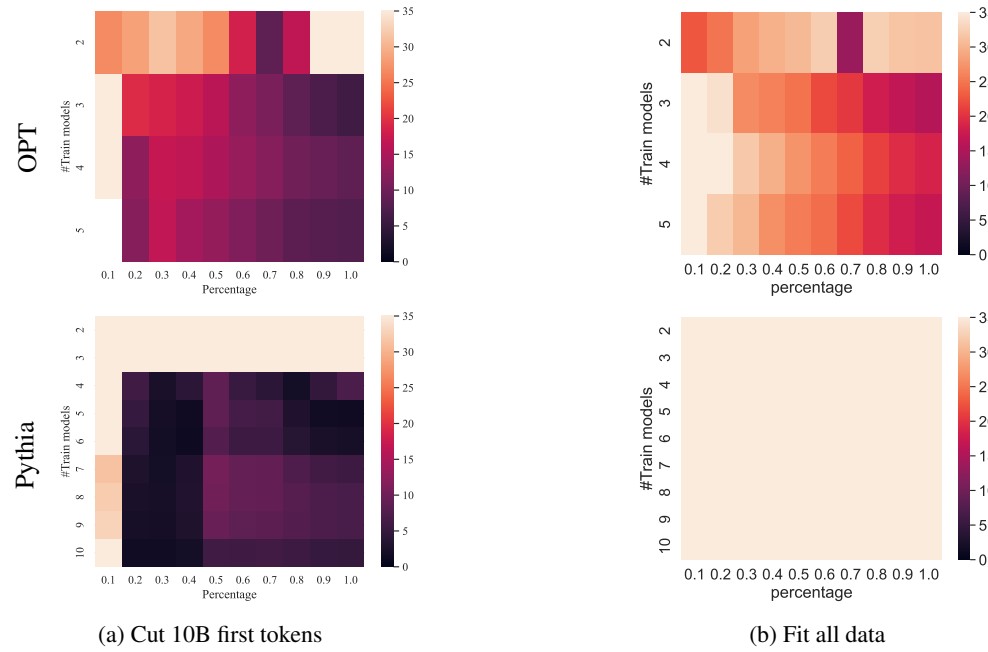

(a) Cut 10B first tokens                                   (b) Fit all data

Figure 5: The effect of fitting with all the training losses and without the beginning 10B tokens seen. Each cell represents the absolute relative error when estimating a scaling law from a given number of models (vertical axis) trained on a given subset of checkpoints from the beginning of training (horizontal axis).

data (as might be suggested by GPT-3 results alone) or only some of it. But it is rarely the case that best estimates are obtained from the end of training alone.

There is, however, a distinctly uninformative phase at the beginning of training, as can be seen in the loss curves (App. B) and noted in the literature (e.g., Chen et al.). We observe that this period is more likely to contain significant spikes or an increase in loss (worse performance) despite additional training. We hence hypothesize this part should always be removed from the scaling law.

Indeed, our experiments depicted in Fig. 5 compare scaling law AREs with and without including models trained on less than 10B tokens in $F$. Evidently, the very beginning of training (often not even reported in logs and graphs) is sometimes harmful to the prediction and is perhaps more noisy. Specifically, we run the same experiments with and without ignoring the first 10B tokens seen. We find that for some models (e.g., OPT and Pythia) the ARE exceeds 15% even when using the whole data, but drops to 4-10% when ignoring those tokens. In preliminary experiments, we found that cutting fewer tokens gave noisier results, and cutting more had a negligible effect.

# 7 HOW BIG A MODEL SHOULD I TRAIN?

*Larger models are better, but not necessary. Mainly, beware of specific models that might give noisy results.*

In Fig. 2 we compare scaling laws when controlling the amount, percentage, or size of the models (2 at a time). We find that choosing models closer in #params to the target model is generally effective (e.g., Fig. 2a, 2c), but the effect is neither strong nor monotonic. For example, in all cases fitting on all $F$ provides one of the lowest ARE. However, in GPT, Gopher and OPT, predicting with the smallest 4 models available is already enough to achieve less than 10% error. In Pythia, the smallest models are not predictive but the rest of the models provide a similar fit. While relying on a larger model is beneficial, predicting many scales up (e.g., the behavior of a $34\times$ larger model in Pythia) is still reliable, especially if accounting for other factors we discuss next.

In fact, training additional, larger models before fitting a scaling law may sometimes decrease accuracy due to increased variance in large model performance—see, for example, Pythia 2.8B in Fig. 1. Unfortunately, it is difficult to identify whether a seed is exceptionally high or low-performing without additional information. For example, cross-validation on $F$ fails to detect it (see App. D).

Instead, this instability can be addressed by accounting for seed variability. A wasteful way to do so would be to train every model several times. A better alternative is to diversify and train each model on as differing hyperparameters (here, seed, #params, #toks) as possible and to maximize the information gained (a common practice in efficiency-coverage scenarios, e.g., Perlitz et al., 2024). Hence, we suggest training more models of differing sizes each accounting for both size and seed changes, rather than training multiple seeds. We further discuss the effects of number of models ($|F|$) in §8.

Selection of #params values to optimize statistical and computational efficiency is a problem we leave for future work. Given the choice of the largest model and the number of models, it is unclear how to space the model sizes, whether linearly, log-scale, or otherwise.

# 8 HOW MANY MODELS ARE NEEDED FOR RELIABLE PREDICTIONS?

*5 models is a safe bet, more would improve the results' robustness. These models can be small.*

We have seen that predicting with larger models and hence extrapolating less yields better results. However, given compute constraints (and additional hardware constraints like memory), practitioners may generally wish to use smaller models when possible. Consider for example Fig. 2b where we compare fitting on 4 models but vary their size. We find that *more* models reduce ARE even without being *bigger* models. As discussed in §7, adding a larger model to a current scaled family serves two goals, it increases the proximity to the predicted model, as well as increases the number of models seen.

We separate the contribution of size and number of models effect. In Fig. 2c, we predict with the largest model being held constant and add (at minimal cost) smaller models. We see again that larger models do benefit predictions. For example, the small models part (left) of the graph indicates large errors (bright). However, we also see again the unwanted effects a single model may have on the overall prediction. Consider for example the figure's diagonal in Pythia. Cells in a diagonal share a group of models and each row adds another one to $F$. Evidently this specific group hurts results, even when larger models are added to $F$. With enough models (bottom of diagonal), the negative decreases. Switching the model (next column) also removes the negative effect. Moreover, across all rows the tendency is never monotonic, implying larger models do not not ensure better predictions.

But in general, we see that increasing the number of models tends to improve prediction. For example, in GPT3 the best predictions are with many models. Perhaps intuitively, adding a larger model and improving both #params and number of models aspects improves quite consistently (Fig. 2b and diagonals of Fig. 2c).

# 9 WHAT PARAMETERS DO I ACTUALLY NEED TO ESTIMATE?

*Scaling laws might have fewer degrees of freedom than described in the literature.*

Assuming we do not try to account for aspects other than #toks and #params (see §10), one might wonder if some of the observed errors come from model misspecification—an incorrect functional form for $\hat{L}$, which (with a small number of exceptions including Caballero et al.) has generally gone uncontested since it was first proposed (Rosenfeld et al.; Hoffmann et al., 2022). Here we specifically evaluate whether scaling laws empirically exhibit fewer degrees of freedom than has been proposed. First, we compute the principal components of the 5 learned parameters and find that 3 components explain 99.49% of the variance between the 5 parameters. Inspection reveals that two of these components tightly couple the pairs of parameters dealing with the same training parameter (#params and #toks). Plotting values of $A$ against $\alpha$ and of $B$ against $\beta$ (Fig. 3), we see a clear linear relationship between these variables despite ther non-linear interaction in Eq. 1. There are a

few exceptions: the Encoder-Decoder model T5-Pile shows a different behavior from the rest of the scaled families, and four additional scaled families show a different relationship between $B$ and $\beta$. In fact, all these families share the common feature that they were trained using multiple passes over a single training set Gadre et al. (2024). The outlier point with $\beta > 4$ is a 70m baseline of Pythia for a continual training intervention experiment (Biderman et al., 2023). Future work may consider different function forms tying some of the parameters or introducing other ones instead.

Another change for the function form that future work should consider is accounting for the learning rate schedule, as our experiments assumed it was negligible. A mismatch between the form and the real dependence might explain the inconsistencies in using the beginning of training. As noted in §6.1, the beginning is not fitting as well as later on, which we also see to some extent in the percentages axis of Fig.4. This might also be expected as previous works did not take the training trajectory (and loss schedule) into account and ignored this data.

## 10 RELATED WORK

This work builds on a large number of recent studies relating scaling law estimation and decision-making about model training. Among the aspects studied are total training costs including inference (Sardana et al.), effects of sophisticated data selection (Sorscher et al., 2022; Ge et al., 2024), training time (Inbar & Sernau, 2024), transfer of learned skills (Hernandez et al., 2021), behavior of models in other modalities (Mikami et al., 2022; Abnar et al.; Alabdulmohsin et al., 2024; Hesslow et al., 2022) mixtures of experts (Ludziejewski et al.), data mixing (Ge et al., 2024), downstream performance (Muennighoff et al., 2024), vocabulary size (Tao et al., 2024), and architecture comparisons (Tay et al., 2023; Poli et al., 2024) including small models (Muckatira et al., 2024) or other phenomena like finetuning (Zhang et al.) and the loss in different positions in the training sequences (Xiong et al., 2024). Especially relevant to our context is Ruan et al. (2024); Maia Polo et al. (2024) that rely on multiple pretraining settings for creating scaling laws that generalize across models or kinds of losses.

Another line of works that can be seen as a scaling law discusses the relation between model width and hyperparameter choices (rather than loss) (Yang et al., 2022; 2021; Blake et al., 2024; Lingle, 2024).

## 11 LIMITATIONS

Our use of ARE as a primary evaluation metric does not distinguish between over-estimation or under-estimation of performance. When using scaling laws to choose between candidate models to train, these error estimates may be unnecessarily conservative (e.g. if both families' laws are biased in the same direction).

Another major limitation in this study is the difficulty of aggregating information across model families. As most published families evaluate models of incomparable scales, often over incomparable ranges, we were unable to produce an informative version of Fig. 2 that aggregated information across all models available, and was thus able to give general recommendations about compute-optimal choice of preliminary experiments.

## 12 DISCUSSION

This paper provides a first study of open questions in the estimation of scaling laws and their relation to large-scale pretraining decisions. We expect that many of these conclusions could be sharpened or extended with the availability of additional information about model training, and we call on other leaders of large-scale training efforts to share training losses and evaluation results from multiple checkpoints during pretraining—even in cases where model parameters themselves cannot be released.

Our findings leave open many important questions, from performing efficient predictions by fitting on many model families to scaling laws of the deltas between a/b test for a change in attribute (e.g. optimizer) or generalize from one a/b test to another, and to other methods of efficiently compare architectures that do not rely on multiple models (e.g. continual learning). In addition, our results in §9 suggest other scaling law parameterizations might better fit data.

| Practical reccomendations | |
|---|---|
| §4 | Set an estimation goal and a budget |
| §5.1 | If budget allows, train the whole model for 30% |
| §A | If extremely constrained, consider predicting with one model |
| §6 | Use all training losses (except the beginning) |
| §7 | Train as big as possible, but limit tokens |
| §7 | Train more models, not only larger |

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

## A  SCALE UP WITH 1 MODEL

We bring errors of data from fitting from a single model on a given percentage of training to the largest model with full training. Scaling is constant and follows the literature (Muennighoff et al., 2024) and the largest model stands as target model (so the bottom line in each figure represents predicting from the beginning of training). In parallel to this paper, an even more efficient work on predicting with 1 model was suggested, and the two should be incorporated (Maia Polo et al., 2024).

## B  LOSS CURVES AND PREDICTIONS

We provide in Fig. 7 graphs of the loss during training of the target models per originating source (e.g., a paper) together with the predictions by using different percentage of the training.

## C  IS SCALING WORKING ONLY UPWARDS?

*No. Small models usually show consistent and predicatable performance.*

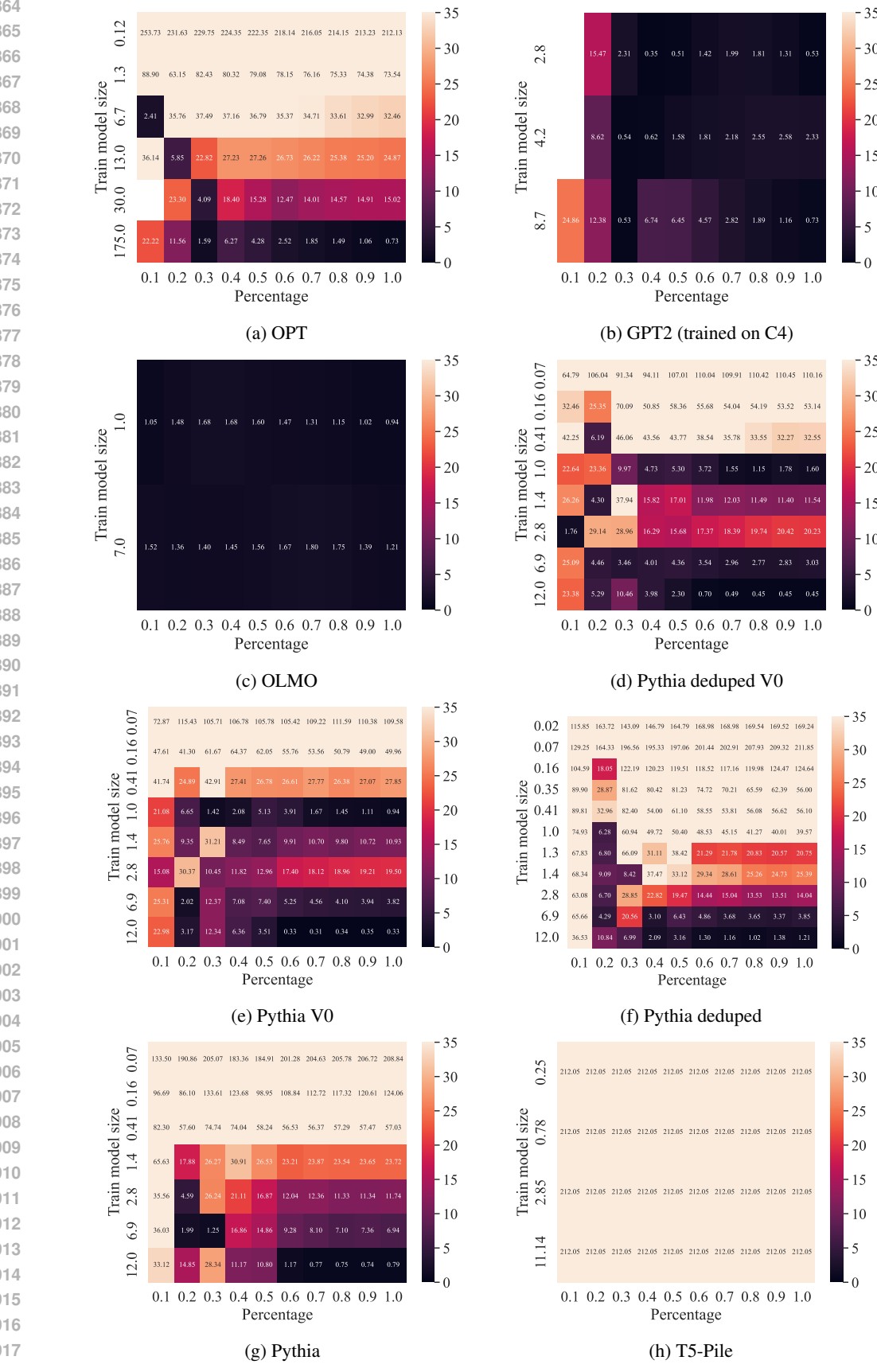

Figure 6: Fitting scaling laws under the assumption that all models scale similarly. Thus, a single model is needed to predict. The last row in each Figure represents predicting a model at the beginning of its training.

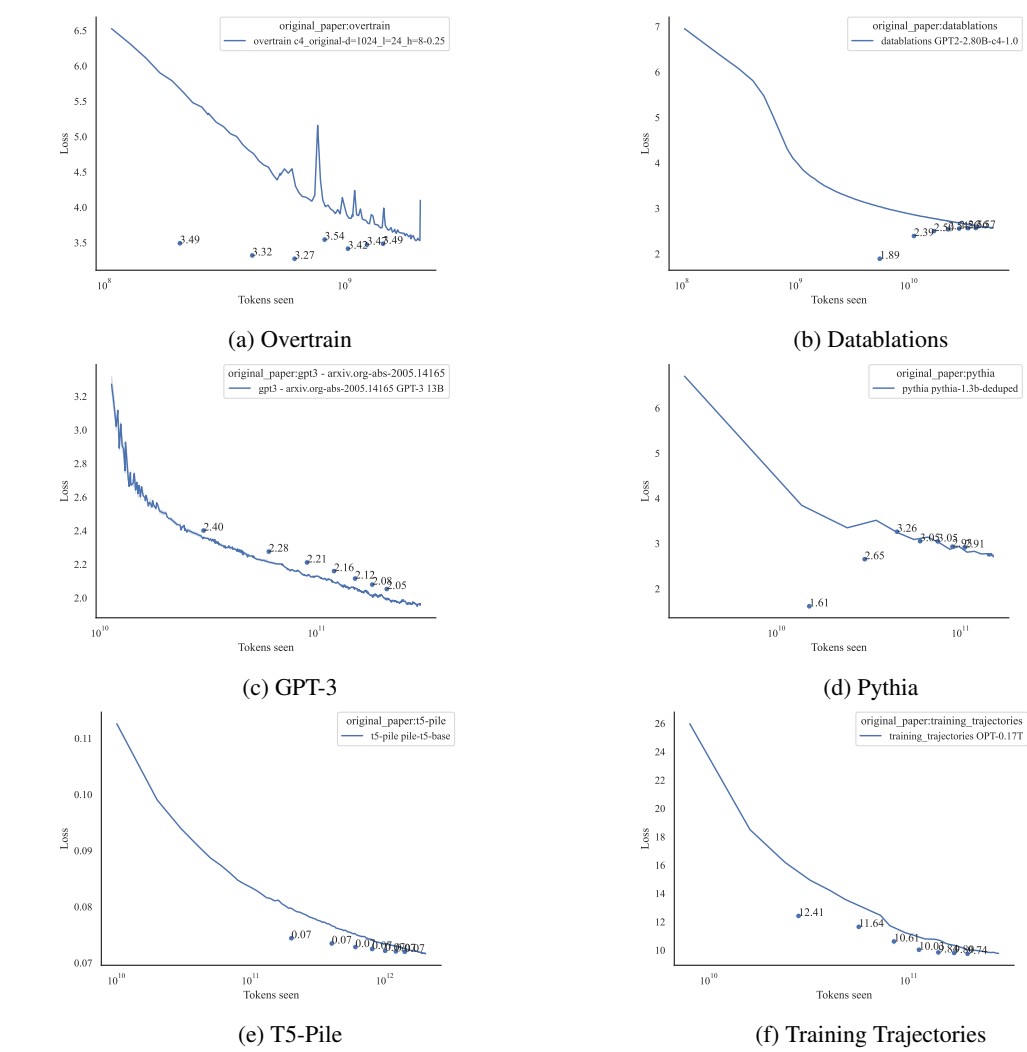

Figure 7: In each figure all losses from a specific source and predictions of the scaling loss with different percentage of the #toks and all models. Predictions are points where the X axis is the available data for prediction and y the prediction value, lines are the actual value. One scaled family per paper was sampled as a representative.

Usually, one does not use a scaling law to extrapolate to a smaller model as one can just train the small model. However, under observational scaling laws, where one wants to research a phenomenon without scaling at all (Ruan et al., 2024; Maia Polo et al., 2024), or when many models were trained and one wishes to create smaller models for various reasons (Hillier et al., 2024; Warstadt et al., 2023), scaling down might prove useful. Moreover, in the context of traditional scaling laws this may act as a baseline. Such an experiment may shed another light on the number of models $|F|$ versus their size #params. If large models are better because they are more stable or otherwise fit laws more robustly, few models will be enough, if the number of models or scale down difference from the prediction, it will show similar behaviour to scaling up. See more in §8.

To test this we reverse the order of models and predict with the largest models the loss on the smallest models. This means that for example in the case of 3 models, we predict the smallest model's loss and fit the scaling law relying on the 3 largest models. As before, we break the results by the percentage of training done and do not reverse it.

As shown in Fig. 8, the number of models plays an important role in fitting well and a minimum of 30-40% of the training is necessary for good fit, more than that often improves further.

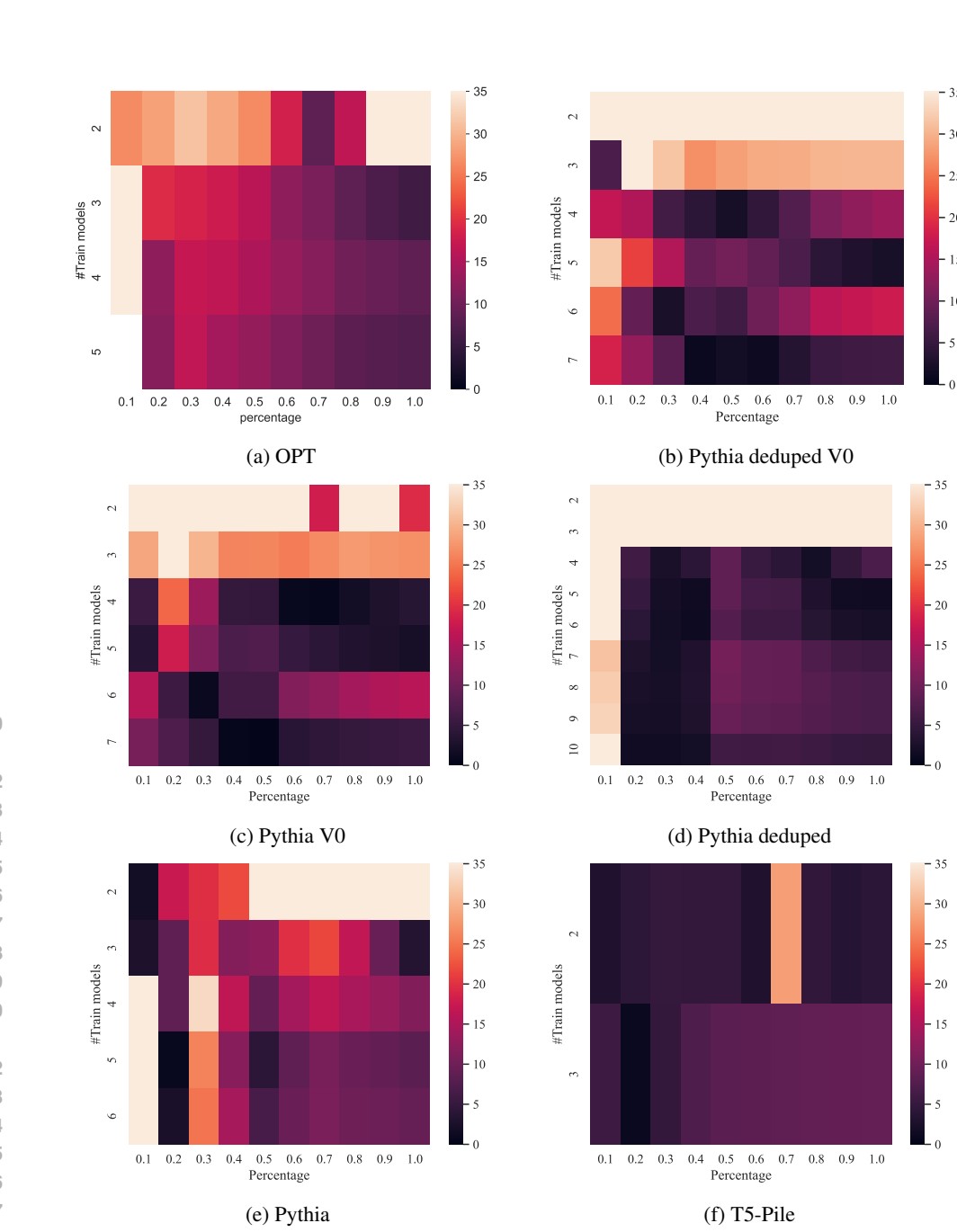

Figure 8: Fitting scaling laws trying to predict the smallest model, with the largest (Y-axis) models trained on a percentage of the data (X-axis).

# D    CAN WE DETECT BAD MODELS TO FIT ON?

*If so, not through cross validataion.*

In §7, we raise the issue of instability of scaling law predictions, with a single model vastly changing
the results. We tried to see if, without knowing the ARE, we could remove bad models from the
prediction. We hypothesized that models that can't predict would mean models that would skew
our predictions when fitted upon. We performed a cross-validation on the #params families in $F$
each time setting the models with most #toks as target ans excluding the #params family from
$F$. Our hypothesis was found to be incorrect. Such cases of hard-to-predict models were found to
indicate that the models left in $F$ are bad predictors and not that the target is very dissimilar (a "bad"
training). In 58% of the cases removing that model from the scaling law created the worst ARE
possible on the actual target, more than removing any other model.

# E    HUBER REPLICATION

Huber loss is sometimes used instead of ARE (Hoffmann et al., 2022). Huber loss is defined as

$$
L_\delta(a) = \begin{cases} \frac{1}{2}a^2 & \text{for } |a| \leq \delta, \\ \delta \cdot \left(|a| - \frac{1}{2}\delta\right), & \text{otherwise.} \end{cases}
\tag{2}
$$

We use $\delta = e10^{-3}$ as done in (Hoffmann et al., 2022). The overall results are similar but for
completeness report them in Fig. 9.

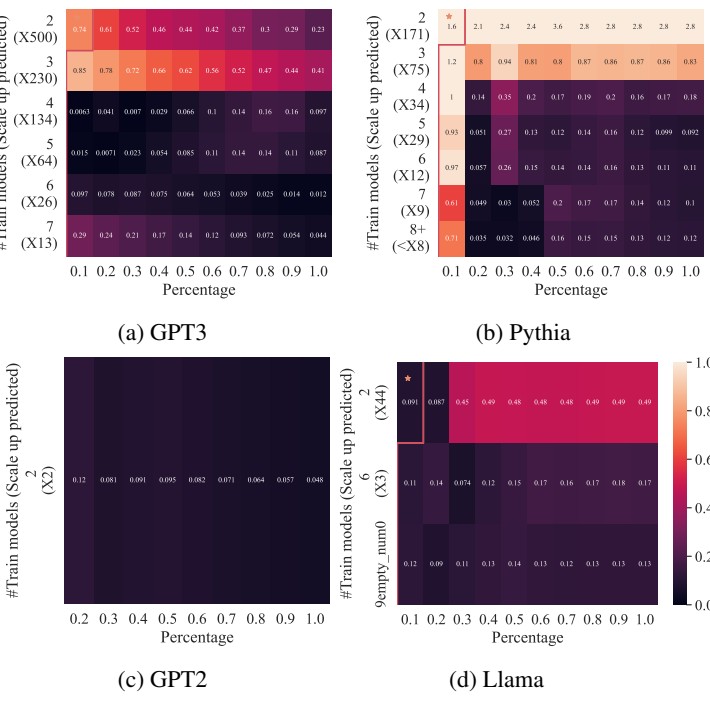

Figure 9: Scaling laws under a Huber loss. The line represents most efficient setting to recieve <0.05.
Comparison of several models given different amount of models and percentages of training.

