# OpenReview forum: "A Hitchhiker's Guide to Scaling Law Estimation"
_ICLR.cc/2025/Conference — Submitted to ICLR 2025_

### Official Review · Reviewer_P9Vb · 2024-10-26

**Soundness:** 3
**Presentation:** 3
**Contribution:** 3
**Rating:** 6
**Confidence:** 3

**Summary:**

This paper conducts extensive experiments to investigate scaling laws in large language models (LLMs). The topic is both relevant and insightful, providing valuable guidance for future experimental design in this area.

**Strengths:**

- The exploration of scaling laws is compelling and relevant, offering insights that can guide experimental decisions in LLM research.
- The paper clearly presents relationships between parameters across various scaling laws (e.g., as illustrated in Figure 3).
- It offers useful findings, such as the preference for extrapolation from a larger number of small, partially trained models rather than a smaller number of large models (as shown in Figure 1).
- Each section concludes with clear takeaways, helping readers understand the implications of each experimental finding.

**Weaknesses:**

- W1: Figures, especially in the appendix (such as Figures 6 and 7), need improvement as some numbers overflow the plot areas.
- W2: Experiments on larger and more recent models, such as LLaMA 3 (70B or above), would strengthen the relevance and impact of the findings.

**Questions:**

- Q1: Could you specify the data collection process in more detail, such as the types of public datasets used and the methods for aggregating them as described in Section 3?
- Q2: Could you add a summarization table to capture the implications of the experiments, making it clearer how readers can apply the findings?

---

> ### Author Response · Authors · 2024-11-20
> **Presentation**
>
> Thanks for the clear review and the show of interest.
> Another thanks for catching the appendix figure state, we revised it.

---

> ### Author Response · Authors · 2024-11-20
> **Model Choice (and llama)**
>
> Regarding models, the newest and biggest model we found was GPT3 (OLMO, Amber and K2 are also new but only offer 1 size so they are less useful for the core experiments). We also wished we could have added llama, however llama is an open weights model, but it did not open its loss throughout training, training logs etc. So we are unable to include it for the lack of data, if you have suggestions for other models that do share validation losses throughout their training and sizes, now or in the future, we are happy to add them to the dataset.

---

> ### Author Response · Authors · 2024-11-20
> **Q1 Data collection**
>
> Q1 We revised the paper to include the information suggested in Section 3, if you have other suggestions we would be happy to hear them.
> For your interest, we elaborate on the way it is done here (and keep the technical details to the paper revision). There are no public datasets for losses sadly, so the collection was done one by one, and parsing manually and uniquely from logs, databases W&B websites etc. To find those, we reached out to the community, searched all the papers of all model releases, talked to people and especially the training dynamics community. For each relevant paper, we parsed whatever it provided, that means some had many sizes, others less so, some had a loss on every few batches and some more etc. Also of course formats, the types of losses used and so on were all very different. Last, on Chinchila and GPT3 that did not share anything beyond the paper, we extracted with a semi automatic tool (line by line) the losses from their graphs and verified the success of the extraction manually.

---

> ### Author Response · Authors · 2024-11-20
> **Q2 summary**
>
> Q2 We added a summary table, a useful idea, thanks. However, it is still initial and we will have something prettier by the camera ready. If you have ideas and suggestions, we are happy to hear them.

---

> > ### Comment · Reviewer_P9Vb · 2024-11-24
> >
> > I appreciate your efforts in data collection and believe the scaling law dataset will be valuable for future research. I will maintain my score.

---

### Official Review · Reviewer_qAoK · 2024-10-30

**Soundness:** 2
**Presentation:** 3
**Contribution:** 2
**Rating:** 3
**Confidence:** 4

**Summary:**

This paper provides a collections of training curves that can be used to investigate different scaling laws formulations. Specifically, the authors collected a large-scale dataset with information on losses and evaluations from 485 models to estimate over 1,000 scaling laws, exploring factors affecting accuracy in scaling law predictions, such as model size, training checkpoints, and family-specific characteristics.

The authors provide a metric named absolute relative error (ARE) to estimate the accuracy of the scaling law fitting, and based on this metric, the authors investigated various factors that may affect the accuracy of scaling law prediction. They have also provided analysis on some important research questions about scaling law.

**Strengths:**

1. A large collection of training curves have been collected. This dataset can be used in further studies of scaling law and also facilitate the study of LLM training dynamics.

2. Some important research questions about fitting scaling laws equations are proposed. Some of these questions are worth further investigating. Such as "if we can do 'transfer learning' with fitting scaling law equations for different experiment settings". Although I think there is some problems of their analysis process.

**Weaknesses:**

1. **The authors ignored an important factor for LLM training: learning rate schedule.**

In fact, the authors have pointed out that previous studies "is based on the assumption that changes in the learning rate schedule" "render losses from intermediate checkpoints uninformative". This rule is regarded to be true in almost all previous studies on scaling laws, including Kaplan et al., Hoffmann et al., and (Porian et al., 2024).

The authors cited two previous studies in line 392-395 to motivate their claim that intermediate checkpoints can be used in fitting scaling law equations. But I have to point out that these two previous studies they have cited do not provide any solid supports about using intermediate checkpoints in scaling law equation fitting. Specifically:

- (Hu et al., 2024) proposed the WSD learning rate scheduler, in which the validation loss decrease gently when the LR is constant and the validation loss decrease sharply as the LR anneals. The validation loss yield by the intermediate checkpoints in this learning rate scheduler do not fit the shape of the scaling law equation in eq.1 at all. In particular, eq.1 can not model the sharp drop of validation loss in the anneal stage. The validation losses obtained from these intermediate checkpoints **can not** be used to fit equations in forms of eq.1.

- (Porian et al., 2024)  found that careful learning rate decay is not essential for the validity of the scaling law. In their studies, they used the **final loss values** yield by the constant LR and cosine LR decay to fit the same form of scaling law equation and found little difference. They did not discussed to use the intermediate checkpoints to fit the scaling law equation.

2. **The definition of ARE is ill formed.**

The author define ARE as the mean prediction error on `max_{#toks}(F_{max}, 0.3)`. This definition implies that we can use eq.1 to predict losses of intermediate checkpoints, in particular the loss values of the last 30% of one intact training process, since the definition of `max_{#toks}(F,q)` "does not distinguish between partially trained models on one hand, and models trained to convergence on a subset of the largest training set used in a family on the other.".

This is simply *wrong*. For example, let's consider a WSD scheduler where the LR remains constant in the first 70% of training and starts to decay in the last 30% of training. The equation fitted using the loss values obtained from the first 70% of steps is not aware of how the LR is going to decay in the last 30% of training. Therefore, the prediction becomes a curve fitting game and provides absolutely no meaningful information.

All the following discussion and analysis of this paper is build upon the calculation of ARE. So if ARE is ill formed. I think we need to question the analysis results obtained based on ARE. It would be better to re-evaluate your key findings using alternative metrics.

**Questions:**

Does the collected dataset contain loss curves obtained from the WSD scheduler? or other learning scheduler?

---

> ### Author Response · Authors · 2024-11-20
> **Mathematical guarantees and prior work**
>
> Thank you for encouraging our directions, the data and the importance of the work.
>
> The main weakness mentioned is that scaling laws on checkpoints are not mathematically guaranteed to work, because of the learning schedules and specifically it was not claimed by previous work. Indeed, the studies we cited do not show intermediate checkpoints aid in scaling law fitting and the training data is by no way mathematically guaranteed to help. As well stated by the reviewer and as we concur (e.g. on Section 9,6 and we have emphasized this point in our revision).
> Nevertheless, we report a new and important *empirical* finding. In Section 6 and the accompanying experiments (e.g., Fig. 4) we showcase that across different models, the data from training is useful, specifically, more useful than ignoring it and relying solely on the final loss. As we noted in Section 9,  accounting for the schedule in the function should be possible and further improve fit, but we leave it for future work, as this is too broad, the loss schedules can take many forms and a general and well motivated approximation merits a paper of its own. In any case, we find across models that it is more beneficial to add this information even without a special function form than ignoring it (as was done before).
>
> We note as you mentioned ARE, that ARE is not an inherent part of our paper but just a common metric used in statistics for such cases, we replicated the results with Huber loss and added the replication to the paper. For example, we still see the benefits from the beginning of training.

---

> ### Author Response · Authors · 2024-11-20
> **WSD in the dataset**
>
> Regarding the question, the data collected comes from many sources, some of which do use WSD (e.g., pythia) and some do not (e.g., datablations scaling laws use cosine), and in some we can’t tell (e.g., GPT3). We have seen empirical evidence that even in models with WSD training the ARE is quite low and benefits from using the methods mentioned.

---

> > ### Comment · Reviewer_qAoK · 2024-11-25
> > **Follow-up about ARE**
> >
> > I think the defect of using the ARE metric to evaluate the scaling law equation is fatal for this paper.
> >
> > Again, as I have mentioned in my review:
> >
> > For a WSD scheduler where the LR remains constant in the first 70% of training and starts to decay in the last 30% of training. There is no point of using the first 70% loss points to predict the last 30% loss points since we don't know how it gonna decay.
> >
> > I want to emphasize one of my major concerns again:
> >
> >  **All the following discussion and analysis of this paper is build upon the calculation of ARE. So if ARE is ill formed (for some of the loss curves). I think we need to question the analysis results obtained based on ARE. It would be better to re-evaluate your key findings using alternative metrics.**
> >
> > I understand the training dynamic of LLM is too complicated to capture. It is OK to constrain the loss curves (as well as all your conclusions) to some commonly used LR schedulers like constant or cosine. For example, you can exclude the loss curves that are trained using WSD or some unknown LR schedulers. But I think such discussions should be highlighted in your paper.

---

> ### Author Response · Authors · 2024-11-25
>
> Do note that we reevaluated as suggested with Huber loss as well. And in initial experiments we also evaluated it on the end of training, and not on 30%, the choice to use 30% just smoothed the end result a bit.
> What would be a suitable metric that you would find acceptable and fix this in your view? As we replicated our results in different settings and in different schedules (WSD or otherwise). You are right that WSD could make a lot of change (and should if you choose it adversarially), practically we find it is not really the case (in reasonable choices made currently). We mention it in the text as well, but will be also happy to elaborate a bit more if the worry is that readers might miss this point.

---

> ### Author Response · Authors · 2024-11-25
>
> Furthermore, per your request I reran the main experiments (scaling law per percentage per model scale for each model family), with a single point (the end of training), so the 30% is definitely not an issue, the huber and ARE results are similar. In fact, by checking manually GPT3 (unknown) and Pythia (WSD) results I couldn't find a cell where this changed in more than 1%.
> If the results of everything were reported with only the last point would that be satisfactory in your opinion?

---

> > ### Comment · Reviewer_qAoK · 2024-12-02
> > **Thanks for the response**
> >
> > Thank you for your clarification. This work has its merit of collecting a useful dataset.
> >
> > However, my augment regarding to the WSD scheduler is not solved. It is true that "WSD could make a lot of change". It is not really an issue only if "in reasonable choices made currently". It will make your work more rigorous and scientific if you can discuss what is reasonable?
> >
> > Moreover, I think it is critical to consider the impact of the learning rate scheduler if you want to use loss points in a training process since the shape of your loss curve differs significantly if different learning rate schedulers are used.
> >
> > If you do not model the impact of the learning rate scheduler, then do not use the loss points within a training process.
> > Typical scaling law equations require *optimal* settings beside $D$ and $N$.
> >
> > I intend to keep my score.

---

### Official Review · Reviewer_1BK7 · 2024-10-31

**Soundness:** 3
**Presentation:** 3
**Contribution:** 3
**Rating:** 6
**Confidence:** 4

**Summary:**

This work attempts to provide a set of guidelines for employing Neural Scaling Laws (NSL) with #of parameters (P) and #of data tokens (N) in training large language models (LLMs). Concretely, the authors perform downstream inference to compute the test loss for a wide range of LLMs, curating a new dataset which can be used for practitioners and theorists who are interested in scaling laws. The authors then define their target model as the largest possible model which has been trained and study the convergence of NSL predictions to the target performance. From analyzing their data, the authors subsequently give general best practices regarding several important topics related to LLM training, including how well can scaling laws be relied upon when fixing all other parameters aside from P and N, whether scaling laws can transfer between different architectures, how many models should one employ to obtain a reliable scaling law, how many training checkpoints should be used and more.

**Strengths:**

I believe the paper has several clear strengths:

1) The topic is very timely and relevant, there is indeed a large body of work dedicated to both empirical and theoretical studies of NSL, and often times reproducing results in this field is very difficult, therefore a contribution which also includes a large dataset is appreciated.
2) The paper is clear and well written, with minor phrasing mistakes.
3) This work collects and releases a very useful public dataset which allows one to analyze scaling laws from 485 pretrained models, and estimate more than 1000 scaling laws. This is a major community contribution which should not be overlooked and encouraged.
4) The authors provide useful best practices for estimating scaling laws in new model families, including some insights which may not be trivial, such as adding checkpoints in the middle of training can lead to a much better estimate of the final test loss scaling.

**Weaknesses:**

In spite of the strengths, the paper has several drawbacks:

**Main weakness:**

The main weakness of this paper, in my opinion, is in its lack of novelty. The main contribution of the paper in my eyes is the curation and publication of data, which is very useful indeed, but I cannot say that the insights gained by this paper are revolutionary or incredibly deep.

**Minor weaknesses:**
1) The paper is entirely empirical, with no real attempt to understand why the rules of thumb given in the paper seem to be correct. This is understandable but unfortunate.
2) The paper focuses only on language models, but scaling laws apply to other modalities and other model parameters. For instance scaling with compute-optimal scaling was not discussed aside from L514-L518.

**Questions:**

My comments/questions are:

1) L25: 'differ scaling behavior' - missing word
2) Figure 2 appears before Figure 1
3) L87-88: "A scaling law estimates the loss of a costly model by training cheaper ones (see Fig. 2) which share a pretraining procedure and differ by some hyperparameters" I'm not
sure I agree with this definition, it might be the correct "empirical scaling law" definition but there are cases where the theory can be derived and there is no concept of really training smaller models, it's more along the lines of studying the behavior of the optimal solution for a model family under a change of dataset size, number of parameters, and compute.
4) L420 - "on all F provides on of the lowest ARE"
5) L526 - "checkpoitns"

---

> ### Author Response · Authors · 2024-11-20
> **Data and novelty**
>
> Thank you for the careful reading, in general your analysis of what we aimed to do as well as what we (would have wished to but) didn’t do, such as theoretical results is accurate, so thanks again.
>
> We are happy you appreciate the data released, it is not always easy to convey the importance of data. Additionally, we do see several points that are novel in this work, of which some findings are surprising for the community and steered long debates when we shared those with researchers of other scaling law papers for feedback (e.g., the fact that more than last checkpoint loss is useful), other points are known to practitioners but unpublished (e.g., to throw the beginning). Specifically, we highlight a non-exhaustive list of new findings from this paper:
> * Training data is useful for scaling laws
> * Beginning of training should be discarded from scaling laws
> * Minimal loss change to expect is 4% and some changes reach up to 50%
> * Different families scale differently, but only slightly and scaling effect from one can be used for another.
> * The number of models has a large effect on the scaling law (even if those are small)
> * We replicate effects like extrapolation error across model families.
>
> The fact that it all seems intuitive is encouraging as it may suggest it is easy to understand (as you also noted in your review), so thank you for saying that as well.

---

> > ### Author Response · Authors · 2024-11-20
> > **Question**
> >
> > Regarding "A scaling law estimates the loss of a costly model by training cheaper ones (see Fig. 2) which share a pretraining procedure and differ by some hyperparameters"  - yes, you are right, maybe we need to rephrase it a bit, we started with a much broader definition that fits also scaling laws in general (e.g. in economics) and eventually chose to go for as specific and as clear and concrete to LLM study to make it useful and easy to understand, in our setting. Maybe we got it too focused now and we can broaden back a bit.

---

> > > ### Comment · Reviewer_1BK7 · 2024-12-01
> > > **Reply to authors**
> > >
> > > Thank you for your replies and additional points, I will maintain my score.

---

### Official Review · Reviewer_qdZ2 · 2024-11-02

**Soundness:** 3
**Presentation:** 2
**Contribution:** 3
**Rating:** 8
**Confidence:** 1

**Summary:**

This paper studies the estimation of scaling law of large language models. The authors build a dataset containing losses and evaluation of pretrained models, and run estimation of scaling laws on this dataset. The authors arrive at some interesting findings that covers different aspects of scaling law estimation, e.g. the reliability of extrapolation, the use of partially trained model, the difference choices of model sizes.

**Strengths:**

1. The authors build and release a dataset that is useful for the reproduction and further analysis of the results.
2. The authors provide a systematic empirical study to the scaling laws. The empirical results in this paper lead to a vast number of implications on scaling law estimation. Some of them are novel to me.

**Weaknesses:**

I am not en expert in this field, and the paper looks overall good to me. My major concern is that readability can be improved. Many of the results and conclusions are not clearly explained, and require experiment results that are scattered across the paper and appendix, e.g., section 5.1. I would encourage the authors to reorganize the experiments and conclusions part of this paper.

**Questions:**

See weaknesses section.

---

> ### Author Response · Authors · 2024-11-20
>
> Thanks for the effort to review our paper, and the accurate summary of the paper and its contributions.  We are also encouraged by the fact that you could explain all those point despite claiming to be from a surrounding area. This is eventually a guide, and should allow people interested in getting into the field in understanding all the analysis and background you described (and perhaps even having the data to do so cheaply). We will revisit your points and others made and make a thorough effort to improve readability, if you have any other suggestions of improving readability and accessibility, we would be happy to hear. For example, do you feel that after reading the paper you are familiar with what a scaling law is and common uses for it? We incorporated some of the suggestions for improved readability, and will also reorganize 5.1 and the connection between each experiment and its results.

---

### Official Review · Reviewer_tqw4 · 2024-11-09

**Soundness:** 3
**Presentation:** 2
**Contribution:** 2
**Rating:** 3
**Confidence:** 4

**Summary:**

This paper focuses on understanding good practices for fitting scaling laws for language models. The paper first collects and releases a large-scale dataset containing losses and downstream evaluations for published pretrained models. Then it uses this dataset to estimate scaling laws and understand the nuance in how to estimate the scaling law.

**Strengths:**

- The paper organizes and releases a dataset covering existing model’s losses and other information for the study of the scaling laws.
- The paper has taken a detailed approach to understand the relationship between different model families’ fitted scaling laws’ similarities, how to estimate the scaling law for a new model using limited amount of training runs.

**Weaknesses:**

The main weaknesses of this paper, from my point of view, are its practical usefulness and its presentation.

- **[Practical usefulness]** Scaling laws are most useful when the target model whose performance to be predicted require much larger compute than the set of smaller models (with varying tokens, parameters, compute) that can be feasibly explored. This paper performs eval only on the largest models within the model family. So it’s not clear to me how big are the set of the second largest models used for scaling law fitting. In contrast, for Llama 3’s scaling law experiments, their small-scale training runs have compute budget up to $10^{22}$ FLOPS but they use the fitted model to guide the allocation of parameter and token to train the 405B model under $3.8 \times 10^{25}$ FLOPS of compute budget.
    - Besides, although I find the problems explored by the authors on how to build on existing scaling laws and use only limited amount of new model runs to fit scaling laws to be interesting, I’m not sure whether such approaches would be seriously considered by practitioners who want to use scaling laws to train large models. As the authors have shown, over certain regimes, not doing the scaling law fitting by generating a lot of small scale model training runs could be less accurate. I would appreciate the authors’ comments on when they think the approaches explored in Section 5 and 6 would be practically useful.
- **[Presentation could be further improved]**
    - **[Notation]** I find some of the notation used in the paper can be changed to improve readability. For example, $\max_{\\# \textrm{params}} (F)$ is used to represent the set of functions $f$ with maximum number of parameters. I find this notation to be a bit unintuitive. The same is true for $\max_{\\#\text{toks}}(F, q)$.
        - In addition, on line 317, page 6, I think $\hat{L}(\cdot \mid F_{\textrm{train}})$ is meant to mean the **loss** of the largest-compute model, but the current notation defines this to be the **function** that uses the largest compute.
    - **[Figures]** In terms of figures, I think the table representation is a bit difficult to interpret, especially the color bar’s choice as well as the isoFLOP contours. Besides, Figure 6 and 7 in the Appendix have overlapping numbers, which make them difficult to read.
    - **[Codebase]** Because the authors mention the database is one of their contributions, the current codebase is a bit poorly documented, which makes me think they are unready to be used as a public benchmark for follow-up works.
    - **[Organization]** Some discussion of Figure 1 (on page 3) is later on discussed on page 9. I think this could potentially be reorganized to improve the flow of the paper.
- Minor:
    - I think it is incorrect to say the intercept $E$ in the scaling law is the “baseline”. Instead it’s actually the best loss this fitted model class can do given infinite compute and parameters.
    - There is a redundant “future work” on line 436.

**Questions:**

- In terms of the equation on line 164, the paper uses squared error as the loss. But Hoﬀmann, 2022 (Chinchilla) uses Huber loss to make the objective more robust to outliers. Can the authors explain why they don’t consider outlier-robust losses?
- The authors mention the intermediate losses could have spikes or increase, making them less useful as regression target. I wonder if the authors have considered applying outlier removal/smoothing of the losses before the scaling law fitting.

---

> ### Author Response · Authors · 2024-11-20
> **Presentation**
>
> Thank you for all the *presentation* suggestions, this is really helpful and important for us to make it more accessible for researchers to read. We addressed most of your suggestions (e.g., typos, overlaps, intercept, etc.) and will address all of them by the camera ready (mainly organize the code and documentation, thanks for checking it out). Regarding figures, we worked, debated and argued a lot over this visualization, highlighting scaling law prediction changes across parameters in the fitting, cost and ARE. Thus, we would be happy to hear suggestions for how this information can be better conveyed.

---

> ### Author Response · Authors · 2024-11-20
> **Usefulness and practicality**
>
> Regarding usefulness, we agree that one should predict the “largest models within the model family” and want to clarify that we do and that the largest model can often be much larger than the ones used for prediction. Specifically, “it’s not clear to me how big are the set of the second largest models” we report on the y axis of the relevant graphs (e.g., Fig. 1)  the scaling up involved, for example, this ranges from x134-x13 in the case of GPT3 with the full parameters. Note that with partial training the Flops required to the most efficient approaches surpass a x1000 save.
> Regarding llama models, the newest and biggest model we managed to extract was GPT3 (OLMO, Amber and K2 are also new but only offer 1 size so they are less useful for the core experiments). We also wished we could have added llama, however llama is an open weights model, but it did not open its loss throughout training, training logs, etc. So we are unable to include it for the lack of data, if you have suggestions for other models that do share validation losses throughout their training and sizes, now or in the future, we are happy to add them to the dataset.
>
> Practicality of Sections 5,6: Section 5 is indeed more of an analysis then a suggestion for a new approach. Specifically, 5.2 set up *baselines*, those are not meant for practical use, but to motivate the rest of our findings and show what are the gains from making a scaling law with smaller models. Similarly 5.1 mainly shows that training your own model is quite costly and scaling laws are often a necessity and offers a slight reduction in costs (x3) if that is useful enough.
> On the other hand, Section 6 is almost always practical, for example, if you want more stability and better fitting of your scaling law at no more cost then you should use more of the data gathered during training, you should not use the very beginning of training etc. Another case for usefulness is if you want to have more efficiency (which you probably do as you are fitting scaling laws for a reason), you might also consider training more models with less data.

---

> ### Author Response · Authors · 2024-11-20
> **Question Huber**
>
> Regarding least squares, the main reason to use it is stability and ease of use. As we note in “Fitting” in 3.2, we did run the original code of chinchilla to predict scaling laws, and it does replicate existing laws to some extent. However, it is very unstable, their original code runs each algorithm with a large grid search of starting points and even then fails to converge on most non-chinchilla scaling examples (chinchilla has many more models trained). Moreover least squares has known optimization methods that run much much faster than gradient based methods such as LBFGS. For us it is somewhat of a practical concern as running thousands of those add up even with caching.
> For evaluation we do not use least square, but we reran the experiments with huber instead of ARE (using the delta hyperparameter from chinchilla), the trends are the same. We added the main figures to Appendix E.

---

> ### Author Response · Authors · 2024-11-20
> **Question spikes**
>
> Regarding “intermediate losses [...] spikes”, smoothing helps for that yes (some of the data is already smoothed and it is showing), and the scaling law fitting itself should also smooth as it won’t be able to fit those together with the rest of the points. The main instabilities though are not the loss spike during training, the instabilities between training runs, that a seed might make one model converge better than another is the largest source of errors. There we tried detecting outlier models by considering the model that is hardest to predict by the rest of the training models. This however did not work (we mention it in App. D).

---

> ### Comment · Reviewer_tqw4 · 2024-11-22
>
> I would like to thank the authors for their response. Upon reading the other reviews and the authors’ rebuttal, I would like to maintain my current score.
>
> Regarding the rebuttal, I’m not sure the Appendix E is complete as it currently ends abruptly with a colon. (Also I would suggest the authors use a different color to mark the changes they make during the rebuttal period).
>
> I think the paper could benefit from some significant rewriting, better notation, and a well-documented codebase, which I think would require a new submission. Concretely, I would suggest
>
> 1. Provide a well-documented code repository and instructions for the code base.
> 2. Spend more space in the paper describing how the dataset was collected. How are the models within each family relate to each other in terms of architecture, learning rate, hyperparameters? This builds up the foundation that enables readers to correctly use your dataset and understand the arguments made in the following sections.
> 3. Use more concrete notations when talking about the training and test set for each of the scaling law’s fitting experiment (especially the customized experiments in Section 5 onwards). So far, some effort has been devoted to describing the test set, but how the training set over which the scaling law is fitted are not presented through rigorous notations. This makes it much harder to exactly understand how each approach is done and evaluated.

---

### Meta-Review · Area_Chair_6NA9 · 2024-12-21

**Metareview:**

In this work, the authors collect and release a large-scale dataset of loss curves and downstream evaluations for pretrained models for use with improving scaling law predictions. Several practical recommendations are proposed, derived from experiments on this dataset. Reviewers were mixed; while many appreciated the development of a new dataset for studying scaling laws, they also took issue with presentation, the limited focus, and

Following the discussion period, my impression of the current manuscript is that it is still premature and requires some additional reworking, particularly with how the results are presented. The collation of a new dataset is of great use to the community, yet relatively little emphasis is placed on its development. I agree with the recommendations of Reviewer tqw4, who suggests that more time is spent documenting the repository, explaining the methodology underpinning the collection of the dataset, and then presenting each finding clearly as a series of statistical studies (what is the training / test set, what is the model, what were the conclusions, hypothesis tests, etc.) rather than a collection of recommendations. This should also help to avoid concerns about the restricted focus on language modelling. More concrete discussion of the nature of the learning rate schedules in the intermediate checkpoints and providing statistical evidence of trends should also address the concerns of Reviewer qAoK.

**Additional Comments On Reviewer Discussion:**

Reviewer tqw4 maintained a negative impression of the work, raising concerns about the utility of the work, and presentation. While some of their initial concerns could be addressed, the reviewer made three recommendations for a second revision of the work, maintaining their score. Reviewer qdZ2 provided a positive, but low confidence, score, commenting on how readability could be improved. Reviewer 1BK7 claimed the work had limited novelty and criticized the depth of the findings due to its empirical nature, maintaining their score after rebuttal. Reviewer qAoK provided a critical review, highlighting perceived flaws in predicting the end of the training curve due to a lack of details on the learning rate schedules. These concerns were not suitably addressed in the rebuttal. Reviewer P9Vb requested that larger models be considered, although the authors stressed that data for larger models are unavailable.

---

### Decision · Program_Chairs · 2025-01-22

Reject